# Quality of Life and Health in Patients with Chronic Periodontitis: A Qualitative Study

**DOI:** 10.3390/ijerph17134895

**Published:** 2020-07-07

**Authors:** Jeehee Pyo, Mina Lee, Minsu Ock, Jihyun Lee

**Affiliations:** 1Department of Preventive Medicine, Ulsan University Hospital, University of Ulsan College of Medicine, Ulsan 44033, Korea; 9201549@uuh.ulsan.kr (J.P.); inyounsy@naver.com (M.L.); 0733822@uuh.ulsan.kr (M.O.); 2Department of Preventive Medicine, University of Ulsan College of Medicine, Seoul 05505, Korea; 3Department of Periodontology, Ulsan University Hospital, University of Ulsan College of Medicine, Ulsan 44033, Korea

**Keywords:** periodontal disease, health-related quality of life, preventive care

## Abstract

Purpose: Periodontal disease causes tooth loss if not treated early, and advanced periodontitis can cause a decline in various oral functions. These results diminish the health-related quality of life (QOL) for various populations. Thus, early detection and management of the disease, as well as a systematic strategy for the prevention of periodontal disease, are necessary. Methods: Adults, 19 years of age or older and diagnosed with chronic gingivitis or chronic periodontitis under the ICD-10 codes, were selected to participate in the study. A total of 20 participants were informed of the purpose of the study and gave consent to participate in in-depth interviews. Results: The treatment of periodontal disease improved health-related QOL and enabled the participants to have positive dental care health behaviors. Furthermore, the participants recognized the severity of periodontal disease and the importance of dental examinations. It enabled them to be aware of the societal need for dental care awareness. Conclusions: This study was an in-depth examination of the health-related QOL of periodontal patients through a qualitative research methodology. We expect that this study will expand research on health-related QOL due to periodontal disease and revitalize the dental health system and practices.

## 1. Introduction

Periodontal disease is a chronic inflammatory condition of the teeth and gums, accompanied by deterioration of the surrounding connective tissue and alveolar bone, and sometimes tooth loss [1]. In most cases, unfortunately, the disease is left untreated as it shows no indication of discomfort during the early stages; detectable symptoms only occur after extensive disease progression. It has been reported that there is a profound association between periodontal disease and systemic diseases as the untreated and neglected periodontal disease could cause cardiovascular disease, risk of premature birth, and other diseases [2,3,4]. Thus, not only is early detection and management of the disease necessary, but a systematic strategy for the prevention of periodontal disease must also be developed [5].

Disease burden created by periodontal disease has been recognized as a global challenge [6,7]. In Korea, the number of patients receiving outpatient treatment for periodontal disease has increased every year; the number of outpatients in 2012 was about eight million, and in 2017, the number exceeded 15 million for the first time [8]. This is the second most common medical condition, with 1 in 3 people in Korea receiving outpatient treatment for periodontal disease. The burden of medical expenses is expected to increase continuously, parallel to the growth of the elderly population; according to total medical care benefit–cost analysis, periodontal disease had the highest share among all diseases with a cost of 1,241,907,934 won (Korean currency), indicating the extreme economic burden of the disease [8].

Periodontal disease causes tooth loss if not treated early, and advanced periodontitis can lead to a decline in chewing ability, word pronunciation, and aesthetic function. These results diminish the health-related quality of life (QOL) for various populations, particularly the elderly, adults, pregnant women, and workers [9,10,11,12,13]. Some primary examples of impacts on QOL include limited food consumption caused by weakened chewing function, development of gastrointestinal disorders and nutritional imbalances, and inability to comfortably converse and interact socially. Pain from periodontal disease can also lead to absenteeism from work and sleep disorders, resulting in economic damages. Thus, evaluating health-related QOL in the treatment and management of periodontal disease is necessary to allow for designing interventions and monitoring improvements related to the disease burden.

However, qualitative studies exploring the life experiences of periodontal patients in depth are insufficient worldwide [14]. In terms of health-related QOL, information from quantitative evaluations does not easily translate into an understanding of the various issues for patients. Further studies are necessary to obtain foundational data that could enhance the health-related QOL of periodontal disease patients through identifying life experiences pertaining to health-related QOL for these patients. Therefore, this study applied a qualitative research method using in-depth interviews with periodontal disease patients to multifariously examine their experiences and clarify their health-related QOL. 

## 2. Materials and Methods 

We conducted in-depth semi-structured interviews with people living with periodontal disease to multifariously understand their health-related QOL.

### 2.1. Research Participants

Adults, 19 years of age or older and diagnosed with chronic gingivitis (K05.1) or chronic periodontitis (K05.3) under the ICD-10 codes, were selected to participate in the study. Among the patients visiting the dental outpatient department in a university clinic, the study participants were chosen for our sample with the assistance of a periodontist. A total of 20 participants were informed of the purpose of the study and gave consent to participate in in-depth interviews. One participant, who was excluded after the initial interview, revealed that they had completed the periodontal disease treatment, needed only one dental exam per year, and had no health issues relating to the study. Therefore, a total of 19 patients were selected as participants. The participants were compensated with 30,000 won (about 25$) for engaging in the study. The specific sociodemographic characteristics of individual participants are shown in Table 1.

### 2.2. Data Collection Method and Process

A designated researcher explained in detail the purpose and specific contents of the study to each participant, and in-depth interviews were conducted with those who gave their consent. In-depth interviews were conducted by two researchers individually. Both were female, one was a nurse and the other was a researcher with extensive experience in qualitative research (such as writing a dissertation for degrees). Participants recognized the interviewer’s job (nurse and researcher) and gender before conducting the in-depth interviews. The semi-structured one-on-one interviews were conducted from 7 January 2019 to 29 March 2019, and were held in the hospital conference room where the participants could talk quietly and comfortably. Each participant was interviewed once. The interview content was evaluated by one preventive medicine professor and one periodontist for appropriateness of questions prior to being used with the participants. A second review was done following the first two participant interviews to finalize the content. The specific questions in the guidelines included: (1) “Did you consistently receive and manage your dental care before your periodontal disease?”; (2) “What symptoms did you experience that made you think you had a gum problem before you came for dental treatment?”; (3) “What past lifestyle habits do you think led to your periodontal disease?”; (4) “What difficulties have you experienced since you developed periodontal disease?”; (5) “In what ways did your QOL change after developing periodontal disease (compared to before)?”; and (6) “What efforts have you made to cure your periodontal disease?”

Prior to the in-depth interviews, the interviewer tried to build rapport through asking the participants about their daily lives. As a result, during the in-depth interviews, the participants naturally mentioned changes in their perceptions of dentistry, changes in QOL due to periodontal disease, and changes in their overall attitudes toward treatments for periodontal disease and others in their responses. A summary of the complete in-depth interview process is shown in Figure 1. 

### 2.3. Analysis Method and Procedure

The analysis team consisted of one preventive medicine specialist, one nurse, and one preventive medicine researcher. The specialist and the researcher had extensive experience in qualitative research, and all three members had studied qualitative research methods.

The first analysis involved segmenting the transcribed data of the in-depth interviews into semantic units in order to understand the meaning of the participants’ responses. Two researchers led the analysis, and the whole panel applied the procedure, as agreed upon by consensus, to derive semantic units after individually analyzing the complete transcribed data analysis. Following the classification of the responses by each researcher, we cross-reviewed the categorization results of all the participants and reached consensus through extensive discussion. Finally, the categorization framework agreed upon by the two leading researchers was reviewed by a preventive medicine specialist, and data saturation was reached when no new semantic units were discovered [15]. A summary of the entire analysis process is shown in Figure 2.

### 2.4. Research Validity

To ensure the validity of this study, we reviewed four criteria: truth value, applicability, consistency, and neutrality, as proposed by Guba and Lincoln [16]. In order to confirm truth value, a categorization result table was presented to one participant to assess the categorization. The reviewer confirmed the authenticity of patients’ experiences as they pertained to the categorization. In addition, one periodontal disease patient, who did not participate in the study but met the criteria for selection of participants, reviewed the categorization to verify the similarity of their experiences of periodontal disease to retain applicability. Neutrality was achieved as the researchers shared preconceived notions of periodontal disease before the start of the study. Moreover, the researchers attempted to exclude any potential preconceived notions generated from the continuous discussion among the researchers during the study. Lastly, consistency was ensured as the entire process of this study was presented in detail, and the results were derived through cross-checking and discussion among researchers with experience in qualitative research.

The study was conducted after approval from the Institutional Review Board (IRB) of Ulsan A Hospital (IRB No: 2018-11-003).

## 3. Results

The life experiences of 19 participants with periodontal disease were reconstructed using in-depth interviews. A total of 899 key results were obtained and each semantic unit was categorized into one of four upper categories: “Interfering Elements for Dental Care”, “Declined Quality of Life Caused by Dental Disease”, “Satisfaction Elements after Treatment of Dental Disease”, and “Improvements for Voluntary Dental Care”. Details are shown in Table 2.

### 3.1. Interfering Elements for Dental Care

#### 3.1.1. Regrets Regarding Previous Dental Treatment

Most participants had been to other dentists before visiting the current university hospital. The participants have had gingival pain for numerous years and have visited dentists for the purpose of treatment. Unfortunately, the dentists at the previous clinics and hospitals did not offer sufficient explanation about their gingival condition and they made unconditional recommendations for tooth extraction. In addition, the previous dentists only provided medicine when the participants had toothaches and did not educate them on gingival care. As a result, the participants did not experience any improvement in gingival pain even after extraction and medication consumption. Thus, they decided to receive treatment from their current dentists at the periodontal department of the university hospital voluntarily or upon the recommendation of an acquaintance.

“It’s painful … When I went to a general hospital, they said they (would) have to pull it out. (Your tooth?) Yeah, like they have to pull it out.”(Participant 4)

“There was pus coming out of my gums. I didn’t get a precise answer when I went to that dentist, not even from another dentist.”(Participant 13)

“If you go to a clinic, they only give you meds when you tell them you are in pain, no treatment like this, but just give you antibiotics and meds when you tell them it aches. So, after taking the meds, it doesn’t get better and it still wobbles … So they said I should just pull it out … Pull this one out and do (dental) implant, pull that one out and (dental) implant”.(Participant 2)

#### 3.1.2. Lack of Dental-Related Knowledge

Some participants were exposed to an environment where little to no attention was given to the needs of dental care for those who had, for example, lived in an impoverished era of war or had challenging living conditions. Nevertheless, they had never been educated on proper tooth-brushing techniques, even though they had visited a dentist since their living conditions had improved. Their lack of knowledge meant that they were unable to actively manage the periodontal disease symptoms, such as pus, swelling, and bleeding, and, as patients, they unconditionally obeyed the doctors’ orders for tooth extraction.

“I left home to live by myself when I went to high school, so that’s when I started to brush my teeth … I mean, I didn’t know that I had to brush my teeth after breakfast. I was just busy eating my meal and went (to school) … I used to live in (the) countryside, so I didn’t do any dental care at all … There was (also) no concept of brushing.” (Participant 3)

“When I brushed my teeth, there was a bit of blood quite often and (I) felt the swelling, but it didn’t really bother me too much, so I brushed it off.” (Participant 17)

“At the first hospital, it was so bad that they simply told me they had to pull all my teeth out in a few years, (there was) nothing (said) about they will do such and such in order to deal with this or that in what way.” (Participant 17)

#### 3.1.3. Obstacles to Dentistry

The unique environment of the dentist was an obstacle that delayed the participants’ dentistry despite their experiencing symptoms of periodontal disease. They also stated that medical staff from different hospitals gave different treatment directions for the same dental condition and displayed considerable variation in the requested bills, which decreased the credibility of professionalism.

“It’s scary after all, and the atmosphere, I guess? That sound and you have to like, open your mouth like this and show everything. How scary these things are (is) the biggest (reason). (Participant 16)

“I am so afraid of the anesthesia, I have a phobia (about that), so I insisted that I will get it (the treatment) from a university hospital … because of the horror and fear, I couldn’t go (to the dentist) and I was under a situation where I couldn’t get an act of courage as well because I am a type that doesn’t get anesthetized that well.” (Participant 6)

“I think the biggest thing is distrust. Because (you worry) if you go (to the dentist) and they might scam you when you got nothing or anything (wrong with your teeth) … You get scared when you go to a hospital because you are sick and they say “If you don’t do this or that, you will get into big trouble”, instead of saying something like “Your condition is blah blah blah and this should be treated in this particular way”. When I hear stories (about hospital visits), it’s all different from every hospital … Frankly, I am (a) bit skeptical a lot of times.” (Participant 17)

### 3.2. Decreased Quality of Life Caused by Dental Disease

#### 3.2.1. Difficulties in Daily Life

The symptoms of periodontal disease that participants experienced were primarily presence of pus, swelling of the gingiva, tooth wobbling, bleeding, halitosis, and toothache in which the gingiva felt like a sponge. These symptoms were exacerbated when they suffered from sleep deprivation or stress. They claimed that it interrupted the daily routine and one participant said that they even felt a threat to their life. 

“My teeth came up like this and when I chewed something on top (of my teeth), they suddenly slid down like chewing a sponge and then came back up. So, I looked at my gum, and (there was) pus, and the inflammation, it came out when you squeeze it. That’s why I went to the hospital … When I brushed my teeth, I felt pus coming out and there was also a lot of bleeding when I did the brushing.”(Participant 3)

“I think sleep and gum (health) (are) very closely connected. On days when I can’t sleep and felt tired, my gums start to bleed immediately and the blood gushes out the next day.” (Participant 12)

“It was almost to the point where I immediately went into convulsion when I hear D of the dentist. Because I had really really bad teeth or gums. I went through extreme suffering that the meaning of life, I mean I couldn’t find the meaning of it … Since I couldn’t chew on it, so I thought about death too.” (Participant 5)

#### 3.2.2. Difficulties in Social Life

Working life and interpersonal relationships are exceptionally significant in the 21st century. Some participants were unable to properly manage their teeth while working and often abandoned their careers due to neglected dental care caused by frequent after-work drinking sessions. Likewise, in terms of interpersonal relations, halitosis and changes in the appearance of teeth affected by periodontal disease were factors that intimidated them.

“I didn’t want to go to work. You know, when your teeth are like that, it’s hard to work … I can’t keep brushing my teeth … During work, around 10 a.m., I get a coffee break and eat some sweets, but I can’t brush my teeth right then. I can brush my teeth after lunch, but it’s hard to do it thoroughly.” (Participant 1)

“Since people have been pointed out (my teeth), I automatically cover my mouth with my hands whenever I meet someone … some people said harsh things like I look fine with my closed mouth, but I look like a monster when I open my mouth.” (Participant 6)

#### 3.2.3. Economic Difficulties

As one participant said, “I spent money, enough to buy a luxury car, to treat periodontal disease”, it is clear that dental treatment costs were overwhelming for them. However, some were not under this financial pressure, as they were satisfied with the cost-effectiveness of the treatment results. Most participants, however, were distressed by the large medical expenses of dental treatment. Notably, one participant shared their experience of receiving illegal dental treatment from a non-medical professional as they had failed to receive tooth scaling due to monetary difficulty. 

“Just for the sake of the smooth sailing of my treatment … I was happy. Happy, and I had faith (in the treatment).” (Participant 5)

“More than 10 million won for (the treatment of) today also. I’m getting one again this time. I am getting it done again because the previous ones were really bad, and this costs about 5 million won, so you feel extreme economic stress.” (Participant 7)

“The cost of expense is too high. I thought it was 2.5 million won as a total, but if the pillar (implant fixture) takes 1 million won, then it would cost me another pretty penny for visiting back and forth.” (Participant 1)

### 3.3. Satisfaction Elements after Treatment of Dental Disease

#### 3.3.1. Positive Change in Daily Life

The participants who went through the periodontal disease treatment experienced various changes in their daily lives. The most significant change was the improvement of QOL resulting from the relief of the periodontal disease symptoms. Specifically, they were able to eat the food they wanted, taste the flavor of food, preserve their teeth without extraction, and sleep soundly due to toothache relief. 

“It (the pain) suddenly decreased, (it did) not just simply decrease, but it got one in a million times smaller … If before if it was 100 points, now it rapidly decreased to 1 to 5 points.” (Participant 10)

“I couldn’t have eaten well before; Now I eat well … Eating almonds was something that I couldn’t imagine, but now I can. Just as an example, I can even eat almonds.” (Participant 3)

“The pain is gone, so I became 100% confident. Say, for the longest time, what would be a better hope than that we, as human beings, could all have our own teeth for the rest of our lives—it is a blessing. It is the ultimate choice to have your own teeth preserved the most for a long time. That’s why I made sure to keep this tooth. Now I live without any feeling of pain at all.” (Participant 12)

#### 3.3.2. Positive Change in Relationship

The participants who felt much discomfort in their interpersonal relationships due to periodontal disease symptoms said their halitosis and aches were reduced when they were treated for periodontal disease. As a result, they stated that the sharpness of pain disappeared, and they were happy to meet and interact with people. One participant used to cover their mouth with their hands all the time due to halitosis, and when the symptoms improved, they mentioned that they were happy to be able to be interviewed and to have a big smile.

“Tooth (health) is happiness … I have the confidence that I can speak like a normal person without pain, and I can do my job feeling better.” (Participant 5)

“I tend to recommend it (the treatment) to people around me … I get to act in more natural ways. As I forced myself to cover it (my mouth) and unnaturally acted like this in the old days, now I can comfortably face people like this, talk like this, and laugh like this.” (Participant 6)

#### 3.3.3. Increased Positive Awareness of Dentistry (Teeth)

Participants who have been receiving dental treatment considered dental care to be one of the five blessings (longevity, wealth, health, love of virtue, and peaceful death) and highly essential, and recommended dental checkups to those around them. In the past, they had vague fears and low confidence in dentists, but they have started to experience high satisfaction and rate dentists as having high credibility because of the improvement of symptoms resulting from meeting their current medical practitioner and gaining understanding of their condition. They were particularly pleased with the departmentalized system and methodical treatment process as they transferred to the university hospital.

“They let me preserve my entire tooth. I felt really appreciative … I could do dozens of interviews like this about teeth. I really went through a lot. But (thanks to) Dr. Lee … After that (the treatment), my world changed upside down. I really went through a lot because of my teeth”. (Participant 13)

“You know, most people, they tend to have this thought of losing quite a bit with unclarity when you go to a private hospital.… We shouldn’t jump to a conclusion, and they are valuable practitioners, but that particular thought can’t escape my head. But since I’ve visited the university hospital, I think the credibility rose to 100%.” (Participant 12)

“I’ve spread the words to the people around me. You can have healthy teeth by visiting the dentist regularly to get dental checkups and care … I should have done this care sooner, but it’s a pity that I couldn’t have done it.”(Participant 14)

### 3.4. Improvement of Health Behavior for Dental Care

All participants voluntarily made an effort to alleviate periodontal disease symptoms. Many of them brushed their teeth three times a day by applying proper brushing techniques learned at the hospital and utilized dental aids such as dental floss and interdental toothbrushes. They were also actively recommending dental examinations not only for themselves but also for their family members based on their own periodontal disease experience. Moreover, they tried to stop smoking and abstain from drinking. Lastly, participants who were passive and depressed because of periodontal disease demonstrated positive thoughts and actively participated in the hospital treatment process.

“I used to brush my teeth for less than one minute, and nowadays I’m brushing over three minutes, not just three minutes, meticulously, I mean more meticulously. It takes longer than washing my face and hair.” (Participant 10)

“I like to use an interdental toothbrush more. That debris in that unreachable spot with a toothbrush is removable with an interdental toothbrush.” (Participant 11)

### 3.5. Improvements for Voluntary Dental Care

#### 3.5.1. The necessity of Publicity for Dental Examination

Despite compulsory routine dental examinations, most participants did not recognize the importance of these examinations. To improve this impression, they mentioned the necessity of active promotion to raise awareness of the need for dental care through the dental examination smartphone application notification service or brochure distribution.

“You can send out a reminder service to the general public so that people can easily look into it. Or distribute an information brochure—like the notification one that you get when you reach a certain age for the free health checkup—so that you can see for yourself and recognize what periodontal disease is like.”(Participant 6)

#### 3.5.2. The Necessity for Publicity about the Severity of Periodontal Disease

The participants were not aware of the severity of periodontal disease until their symptoms became severe. In addition, even though the symptoms were acknowledged, they were inclined to endure pain without treatment. This behavior resulted from ignorance and fear of the severity of periodontal disease, which requires urgent publicity, according to the participants.

“Unless it (symptom) becomes serious like mine, say you’re just a little uncomfortable, then a lot, I mean most of the people don’t go (to the dentist).…(When you publicize it,) I think it will be a good idea for the Health Insurance Corporation to include a simple guidebook to help you learn about the seriousness, such as what is periodontal disease and what other organs will be affected by periodontal disease.” (Participant 6)

## 4. Discussion

This study analyzed the in-depth interviews of 19 patients with periodontal disease to examine the health-related QOL for those with periodontal disease in various aspects and to investigate the experiences related to periodontal disease. The analysis results were summarized into four categories: “Interfering Elements for Dental Care”, “Decreased Quality of Life Caused by Dental Disease”, “Satisfaction Elements after Treatment of Dental Disease”, and “Improvements for Voluntary Dental Care”. In detail, the patients described regretting experiences of previous dental treatment, dental care management failure due to the lack of dental-related knowledge, experiencing difficulties in daily life and social life, as well as economic burden resulting from toothaches, bleeding, halitosis, and others symptoms of periodontal disease. The treatment of periodontal disease, however, improved their health-related QOL and enabled the participants to embrace positive dental health behaviors. Furthermore, they recognized the severity of periodontal disease and the importance of dental examinations. Proper treatment enabled them to be aware of the need for a societal effort toward dental care awareness.

The foremost significance of this study was that a qualitative research methodology was used to confirm the experiences of periodontal patients in health-related QOL. Periodontal disease is one of the elements that can diminish QOL as related to health. Although precedent studies have been conducted investigating health-related QOL on periodontal patients [1,2,3,4,5,8,9,10], they struggled to comprehensively identify the all-encompassing challenges that the patients face. Health-related QOL should expansively encompass the various dimensions of health, and it is influenced by several factors such as demographic characteristics, depression, fatigue, and personal, familial, and social relationships. Moreover, health-related QOL needs to capture more than the experience of the moment—it must include the conditions before and after treatment and the processes of change. The reality of the current practice environment only allows dental professionals a view of a small portion of a patient’s life as the treatment involves a short encounter between the medical staff and the patient [17]. Thus, to accurately assess health-related QOL before and after treatment for patients with periodontal disease, a qualitative research methodology is required for the in-depth and comprehensive exploration of patients with periodontal disease.

Qualitative research, which enables interpretation of patients’ subjective experiences, can contribute to revealing overlooked or tricky details of conditions analyzed by quantitative studies by providing analyses of content from individual participant interviews. The term ‘subjective’ here indicates that the health condition is significant from patients’ perspective and that interpretation of their subjective experiences could be applied as an essential means of assessing and explaining the consequences of the disease [18].

This qualitative study endeavored to focus on the stories of periodontal patients and thoroughly observe their experiences of life with periodontal disease. The results from this study also demonstrated that not only the treatment of periodontal disease should be highlighted, but also efforts to improve the health-related QOL for those with periodontal disease from a social perspective.

Another primary aspect of this study was that the results confirmed and reiterated the need for persistent management of periodontal disease. In particular, periodontal disease patients, after vigorous periodontal treatment, should receive regular maintenance to prevent disease recurrence. For patients, who previously received only low-quality treatments without maintenance, the condition of teeth rapidly deteriorates starting around the age of 40. However, a combination of high-quality treatments and maintenance can dramatically improve the longevity of teeth. According to the literature on supportive periodontal therapy (SPT) or regular maintenance over five years, the average rate for tooth loss per patient per year is 0.01–0.31 [18,19,20]. Several studies have reported that regular visitation to the dentist and periodontal maintenance reduces tooth loss and loss of adhesion; therefore, there is no existing controversy over the need for periodontal treatment [21,22]. The participants in the study act as living witnesses who have testified that the management of periodontal disease is paramount. For the regular management of periodontal disease patients, establishment of an institutional system like that in place for managing regular visits to physicians is required.

In order to prevent periodontal disease, there needs to be emphasis on dental health as well as positive habitual behaviors for prevention and management. As most of the participants accentuated, to accomplish this idea, the importance of prevention of periodontal disease and the seriousness of the disease should be promoted. It is essential to publicize the decreased QOL caused by dental disease in terms of daily life, social life, and economic aspects. In addition, it is necessary to ensure that visits to the dentist are mandatory for preventive purposes, rather than only when symptoms occur. The greatest need regarding the current dental care system in South Korea is the need to teach and promote the practice of high-quality preventive care. The proportion of the population who regularly visit dentists for preventive purposes is 90% of those between 65 and 84 years old in Sweden. Unfortunately, in Korea, unmet needs persist, and the dental examination rate is low [23,24,25,26,27]. The importance of dental health awareness has not been perceived as a social convention and is vastly different from the status quo in countries with advanced dental care systems [28]. As an eye fundus examination is a part of the process for the prevention of complications when a patient is diagnosed with diabetes, it is necessary to create opportunities to manage periodontal disease by normalizing practices such as receiving dental examinations.

Patients must be aware of the importance of dental maintenance or have the motivation to visit dentists to receive regular dental care. Similar to the participants in this study, patients who are consistently involved in dental care feel proud of and recognize the benefits of maintaining high levels of dental hygiene. In other words, a desire to maintain a healthy oral cavity and enjoy the process would become the driving force for long-term planning for regular visits to the dentists. Furthermore, self-care based on professional brushing training has been shown to be even more important than maintenance several times a year. Customized dental health education programs focused on individual characteristics would be more appropriate for behavioral changes than the standard ones.

As the participants of this study indicated, in the past, dental care has been inconvenient; therefore, it is necessary to enhance the quality of dental care. In Korea, unmet needs are being effectively reduced as result of economic changes, such as improvements to the insurance system, but there is still distrust in the quality of medical care, especially the quality of care experienced at the clinic level of medical institutions [29]. Participants’ dissatisfaction with treatment was due to receiving insufficient explanations of dental conditions, undergoing obligatory tooth extractions, and lacking education on dental maintenance. These elements caused them to doubt dentists and led them to make infrequent visits for dental care. It is expected that the quality of dental care will increase due to the decision to apply the Healthcare Benefit Quality Assessment, which evaluates the cost-effectiveness of the overall care provided by the health insurance, to dental health (it was previously only used in the medical field) [30]. However, improving the quality of dental care will still be a prerequisite to improving access to dental care.

The limitation of this study was that, as the participants were recruited from the general hospital (university hospital), there was a limit in reflecting the experience and perception of patients who are mainly treated in dental clinics. The limitation may have resulted in a more evident demonstration of negative experiences with dental clinics. We suggest that future studies be conducted with periodontal patients who are treated by dental clinics.

## 5. Conclusions

This study was an in-depth examination of the health-related QOL of periodontal patients through a qualitative research methodology. The experiences of periodontal disease identified by this study can not only help to assess the adequacy of the current dental health-related QOL assessment tools, but also to recognize unmet needs regarding periodontal disease and, ultimately, to raise the awareness of periodontal disease among the general public. Based on this research, we expect that research on health-related QOL on periodontal disease would expand and revitalize the dental health system and practices.

## Figures and Tables

**Figure 1 ijerph-17-04895-f001:**
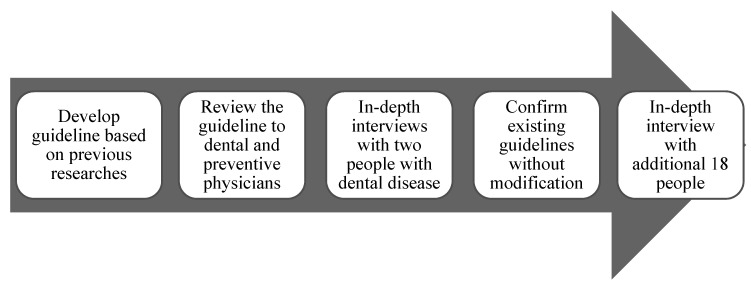
In-depth interview paradigm.

**Figure 2 ijerph-17-04895-f002:**
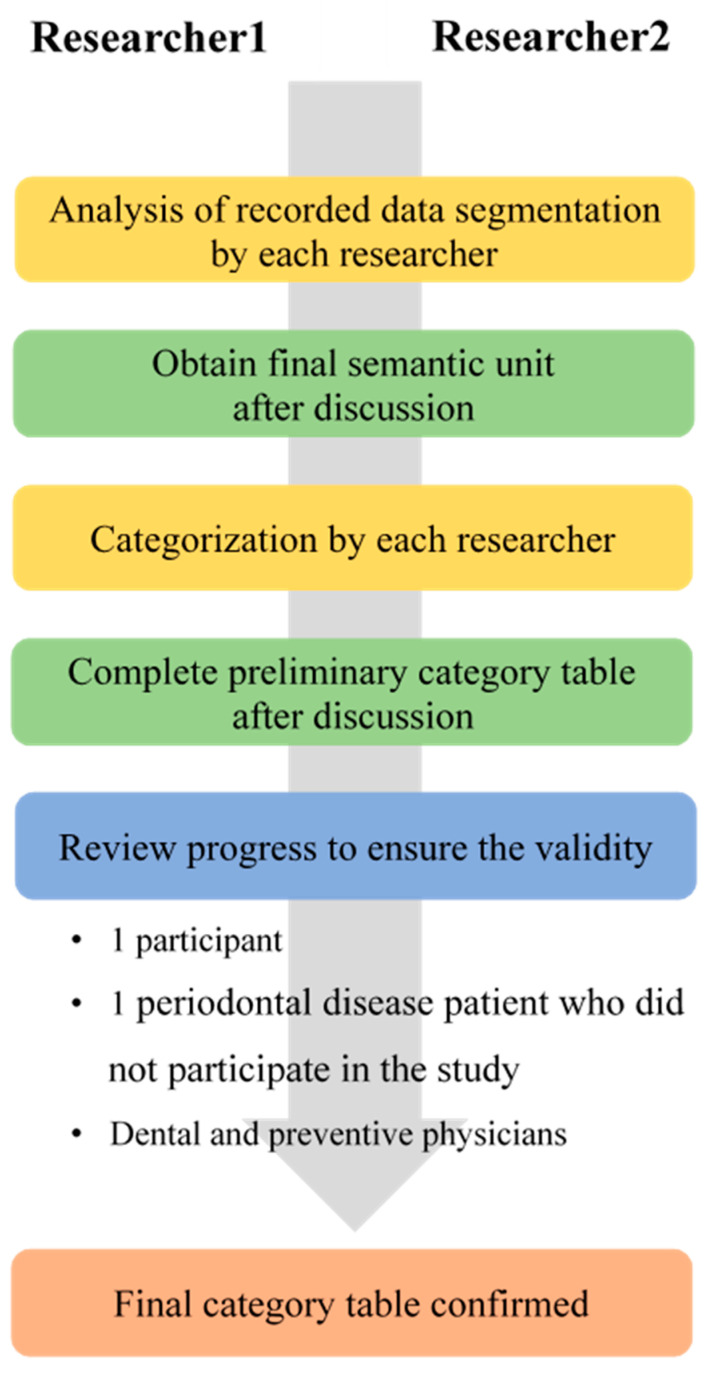
Analysis paradigm.

**Table 1 ijerph-17-04895-t001:** Sociodemographic characteristics of participants.

	Gender	Age	Education	Occupation	Other Diseases/Surgery Experiences	Chronic Dental Disease Treatment Period	Monthly Income (Thousand Won)
1	Female	The 50s	High School	Housewife	-	4 Years	2000
2	Female	The 50s	Elementary School	Housewife	-	4 Years	2000
3	Male	The 50s	4-Year University	Entrepreneur	Hypertension, Diabetes	10 Years	6000
4	Male	The 60s	4-Year University	Security Guard	Colon Polyp	2 Years	1500
5	Female	The 50s	4-Year University	Homeschool Teacher	-	1 Year	2500
6	Female	The 50s	High School	Housewife	Hypertension	3 Years	4500
7	Female	The 60s	4-Year University	Housewife	-	2 Years	6000
8	Female	The 50s	High School	Housewife	Hypertension	1 Year	5000
9	Female	The 40s	4-Year University	Housewife	-	2 Years	7000
10	Male	The 40s	2-Year University	Corporate Worker	-	3 Years	2500
11	Female	The 40s	4-Year University	Nurse	-	1 Year	2500
12	Female	The 60s	4-Year University	Housewife	Arthritis	8 Months	8000
13	Female	The 60s	High School	Housewife	Hip Joint Surgery	6 Years	3000
14	Male	The 50s	4-Year University	Corporate Worker	Diabetes	20 Years	6000
15	Male	The 50s	4-Year University	Corporate Worker	-	10 Years	7000
16	Female	The 50s	4-Year University	Freelancer	-	2 Years	-
17	Male	The 40s	High School	Corporate Worker	Diabetes	2 Years	4000
18	Male	The 50s	High School	Corporate Worker	Kidney Transplant	4 Years	8000
19	Female	The 50s	4-Year University	Nurse	-	17 Years	5000

**Table 2 ijerph-17-04895-t002:** Categorization results.

Upper Category	Subcategory
Interfering Element for Dental Care	Regrets of Previous Dental Treatment
	Lack of Dental-Related Knowledge
	Obstacles to Dentistry
Declined Quality of Life Caused by Dental Disease	Difficulties in Daily Life
	Difficulties in Social Life
	Economic Difficulties
Satisfaction Elements after Treatment of Dental Disease	Positive Change in Daily Life
	Positive Change in Relationship
	Increased Positive Awareness of Dentistry (Teeth)
	Improvement of Health Behavior for Dental Care
Improvements for Voluntary Dental Care	The Necessity of Publicity for Dental Examination
	The Necessity for Publicity about the Severity of Periodontal Disease

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
