# Peer review of "Quality of Life and Health in Patients with Chronic Periodontitis: A Qualitative Study"

_ijerph, 2020, doi:10.3390/ijerph17134895_

Round 1

Reviewer 1 Report

The Authors must be commended for their efforts.

This is a very interesting piece of research using methods that are not widely used in dentistry.

  • Is it true that they received 30.000 won, i.e. 15 times their monthly salary, or a typo?
  • In table two please make clear where the subcategories belong to.
  • Why did the Authors chose the interview approach and did not use the OHIP-CP questionnaire which is designed purely for perio?

Author Response

Point 1: The Authors must be commended for their efforts. This is a very interesting piece of research using methods that are not widely used in dentistry. Is it true that they received 30.000 won, i.e. 15 times their monthly salary, or a typo?

Response 1: Thank you for your valuable suggestion. The 30,000 won paid as a reward is about 25 dollars. I think this is an appropriate amount, not a lot, considering that the monthly salary in Korea is usually 2 million won. To help you understand the contents, we added dollars in ‘Research Participants’ section.

Point 2: In table two please make clear where the subcategories belong to.

Response 2: Table 2 has been modified to make it easier to recognize.

Point 3: Why did the Authors chose the interview approach and did not use the OHIP-CP questionnaire which is designed purely for perio?

Response 3: The OHIP-CP questionnaire has some limitations. For example, the sensitivity and responsiveness of the OHIP-CP were not examined.

Reviewer 2 Report

The manuscript addresses an interesting topic for the journal's readers. However, it is possible to find several published articles on the relationship between periodontitis and quality of life. The manuscript adopted a qualitative methodology, and this is an unusual and relevant feature. However, for the application of this type of methodology, the sampling units must be homogeneous in variables such as age, sex, education, time of treatment of the disease, and the presence of comorbidities. The present study presents a sample with characteristics variability, compromising the analysis and inference of the data.

Author Response

Response : In my opinion it's homogeneous. The purpose of our research is to explore the life experiences of periodontal patients. Therefore, the gender and age group of the participants may be different, but they all have the commonality of periodontal disease. Differences such as gender and age are thought to be factors that further enrich their experiences as periodontal patients. Therefore, the gender and age group of the participants may be different, but they all have the commonality of periodontal disease. Differences such as gender and age are thought to be factors that further enrich their experiences as periodontal patients.

Reviewer 3 Report

Introduction:

Since the second paragraph defines periodontal disease and its natural history the reviewer suggests authors to rearrange the order by making it the first paragraph.

Results:

Table 1 does not show the exact age/age group of the participants. Although the methods indicate that the age of the participants as 19 years or older according to Table 1 all participants aged 40 years or above. 

Author Response

Point 1: Introduction_Since the second paragraph defines periodontal disease and its natural history the reviewer suggests authors to rearrange the order by making it the first paragraph

Response 1: Thank you for your valuable suggestion. I rearranged the first paragraph and the second paragraph. In addition, the references were revised according to the order.

Point 2: Results_Table 1 does not show the exact age/age group of the participants. Although the methods indicate that the age of the participants as 19 years or older according to Table 1 all participants aged 40 years or above.

Response 2: We set the criteria for selecting participants as adults aged 19 or older. However, as mentioned in Introduction section, the symptoms of periodontal disease can be recognized after a considerable period of progress. That's why participants are mostly in their 40s and older. This is the phenomenon itself that we cannot choose from.

Reviewer 4 Report

[Suggestions]
The referee is unable to find any scientific data in the Tables 1 & 2, Figures 1 & 2, as well as in the Results.

Because the referee is not familiar with the methods of the study of in-depth semi-structured interviews, the referee is unable to find any scientific merit in the manuscript. Is this suitable to the publication of Int. J. Environ. Res. Public Health???

Minor:
L. 66-67: "using similar methods as other research on chronic diseases...."
The authors need to cite the appropriate references.

[Typographical errors]
L. 5-7: The authors need to check their affiliations.

L. 515-529: Delete these parts from the manuscript.

Author Response

Point 1: The referee is unable to find any scientific data in the Tables 1 & 2, Figures 1 & 2, as well as in the Results. Because the referee is not familiar with the methods of the study of in-depth semi-structured interviews, the referee is unable to find any scientific merit in the manuscript. Is this suitable to the publication of Int. J. Environ. Res. Public Health???

Response 1: This research multifariously examined periodontal disease patients’ experiences and clarified their health-related QOL via qualitative research methodology. This methodology strives to present in-depth narratives and stories of the lesser number of participants. Also, the methodology studies participants from one person to more than dozens as there is no definite rule on the sample size of the methodology—qualitative research, like ours, endeavors to explore the essence of the stories of participants. On the contrary, quantitative research attempts to generalize a phenomenon through a study with the high-volume of participants. We would like to ask you to consider the differences between the research methodologies.

Point 2: Minor_L. 66-67: "using similar methods as other research on chronic diseases...."

The authors need to cite the appropriate references.

Response 2: The sentence was intended to emphasize that the study used qualitative research methodology and did not mean that it used the methodology of a particular research. We deleted the sentence that could cause misunderstanding.

Point 3: [Typographical errors]  L. 5-7: The authors need to check their affiliations. L. 515-529: Delete these parts from the manuscript.

Response 3: We added authors’ affiliations.

Reviewer 5 Report

Dear Authors,

Congratulation for the study, wihch was probably a hard and difficult work but unfortunately, conclusions are obvious and now well-known...

In addition to the face-to face interview (not anonymous…) it would be interesting to add a questionnaire (anonymous). The major subject dealing with the profound association between periodontal diseases and systemic diseases, is not evoked: no conclusions about those important part of the prevention of systemic diseases.

Introduction must be improve on periodontal diseases subject, and references have to be increase.

Best regards

Author Response

Point 1: Dear Authors, Congratulation for the study, which was probably a hard and difficult work but unfortunately, conclusions are obvious and now well-known...

In addition to the face-to face interview (not anonymous…) it would be interesting to add a questionnaire (anonymous). The major subject dealing with the profound association between periodontal diseases and systemic diseases, is not evoked: no conclusions about those important part of the prevention of systemic diseases.

Response 1: Thank you for your valuable suggestion. This study is meaningful in that it revealed in-depth life experiences of periodontal patients. The study highlighted the importance of preventing periodontal disease and the need for the importance of oral health. Also mentioned was the need to improve the quality of dental care, which was revealed through in-depth interviews with participants. The results written through the language of the participants will reveal specific and rich factors. These results cannot be derived simply as being expressed in numbers. We tried to grasp the meaning of the experience deeply through qualitative research. Scientific research involves systematic and empirical exploration based on data. The characteristics of qualitative research can be seen as scientific research. We have written the content (necessity of qualitative research, significance through this research, etc.) in Introduction and Discussion section.

Point 2: Introduction must be improve on periodontal diseases subject, and references have to be increase.

Response 2:  Our study is a qualitative study on the quality of life of patients with periodontal disease and is shown in Table 1. I added references related to periodontal diseases and systemic diseases.

Reviewer 6 Report

The research team is congratulated for the novelty of the subject of study and the rigorous methodology applied and correctly described in the manuscript submitted. The results are correctly established and discussed.
Some minor aspects to be improved are presented below:

  • It would be interesting to describe in more detail how the study participants were selected.
  • I should have provided more characteristics about the interviewer: gender, experience,What did the participants know about the interviewer? Was a relationship established prior to study commencement? 
  • It would be interesting to include whether audio and/or video recording of the interviews was made, specifying the duration of the interviews, as well as detailing whether there were more people present than the interviewer and the participant. Information is also missing if field notes were taken.
  • Finally, Was any software used to manage the data?

Author Response

Point 1: The research team is congratulated for the novelty of the subject of study and the rigorous methodology applied and correctly described in the manuscript submitted. The results are correctly established and discussed.

Some minor aspects to be improved are presented below:

It would be interesting to describe in more detail how the study participants were selected.

Response 1: I reviewed the content to further describe it, but it was already sufficiently described as true. After setting the criteria for selecting participants, we proceeded with recommendations from the periodontist.

Point 2: I should have provided more characteristics about the interviewer: gender, experience, What did the participants know about the interviewer? Was a relationship established prior to study commencement?

Response 2: The content is described in the data collection method and process.

  1. In-depth interviews were conducted by two researchers individually. Both are female, one is a nurse, and the other is a researcher with extensive experience in qualitative research (such as writing a dissertation for degrees). Participants recognized the interviewer's job (nurse, researcher) and gender before conducting in-depth interviews.
  2. Prior to the in-depth interview, the interviewer tried to form a rappo through asking the participants about their daily lives.

Point 3: It would be interesting to include whether audio and/or video recording of the interviews was made, specifying the duration of the interviews, as well as detailing whether there were more people present than the interviewer and the participant. Information is also missing if field notes were taken.

Response 3: It is written that the recorded data was copied and used in the 'Analysis Method and procedure' section. Likewise, the content of the interview period and the type of interview (one-on-one with researchers and participants) are described in the 'Data Collection Methods and Process' section. Field notes were written, but they were not described separately due to their low utilization.

Point 4: Finally, Was any software used to manage the data?

Response 4: We did not use software (like NVivo). It was analyzed according to the qualitative research analysis guidelines, and the details were described in the analysis process.

Round 2

Reviewer 2 Report

The results of qualitative studies show inference only the sample evaluated and, therefore, their characteristics must be homogeneous to enable less likelihood of bias.

Characteristics such as sex and, mainly, age, influence life experiences that can determine the appearance of periodontal disease.

Besides, some studies have already reported the relationship between periodontal disease and quality of life.
1) Impact of periodontal disease on quality of life: a systematic review. Ferreita et al. 2017. Doi: 10.1111/jre.12436
2) Periodontal disease among older people and its impact on oral health-related quality of life. Kato et al., 2018. Doi: 10.1111/ger.12363
3) Relationship of periodontal disease and domains of oral health-related quality of life. Masood et al. 2019. Doi: 10.1111/jcpe.13072

Author Response

 Thank you for your valuable suggestion. I want to emphasize once again. The purpose of our research is to explore the life experiences of periodontal patients. Therefore, the gender and age group of the participants may be different, but they all have the commonality of periodontal disease. Differences such as gender and age are thought to be factors that further enrich their experiences as periodontal patients. Therefore, the gender and age group of the participants may be different, but they all have the commonality of periodontal disease. Differences such as gender and age are thought to be factors that further enrich their experiences as periodontal patients.

Reviewer 4 Report

Delete the References#31 to #38 from the manuscript.

Author Response

I deleted the References #31 to #38.

Reviewer 5 Report

Dear Author, thank you for your responses, unfortunately: no new informations. The design had to be improve. Thank you for the revised introduction. This study is however interesting.

bests regards 

Author Response

Thank you for your valuable suggestion. Policies for promoting dental healthcare can be developed based on the findings of this study.